# Measurement validity of an electronic training device to assess breathing characteristics during inspiratory muscle training in patients with weaning difficulties

Marine Van Hollebeke[1,2], Diego Poddighe[1,2], Tin Gojevic[1], Beatrix Clerckx[2], Jan Muller[2,3], Greet Hermans[2,3], Rik Gosselink[1,2], Daniel Langer[1,2]*

1 Faculty of Movement and Rehabilitation Sciences, Department of Rehabilitation Sciences, Research Group for Rehabilitation in Internal Disorders, KU Leuven, Leuven, Belgium, 2 Department of Intensive Care Medicine, University Hospitals Leuven, Leuven, Belgium, 3 Division of Cellular and Molecular Medicine, Laboratory of Intensive Care Medicine, KU Leuven, Leuven, Belgium

* Daniel.langer@kuleuven.be

## Abstract

Inspiratory muscle training (IMT) improves respiratory muscle function and might enhance weaning outcomes in patients with weaning difficulties. An electronic inspiratory loading device provides valid, automatically processed information on breathing characteristics during IMT sessions. Adherence to and quality of IMT, as reflected by work of breathing and power generated by inspiratory muscles, are related to improvements in inspiratory muscle function in patients with chronic obstructive pulmonary disease. The aim of this study was to investigate the validity of an electronic training device to assess and provide real-time feedback on breathing characteristics during inspiratory muscle training (IMT) in patient with weaning difficulties. Patients with weaning difficulties performed daily IMT sessions against a tapered flow-resistive load of approximately 30 to 50% of the patient's maximal inspiratory pressure. Airflow and airway pressure measurements were simultaneously collected with the training device (POWERbreatheKH2, POWERbreathe International Ltd, UK) and a portable spirometer (reference device, Pocket-Spiro USB/BT100, M.E.C, Belgium). Breath by breath analysis of 1002 breaths of 27 training sessions (n = 13) against a mean load of 46 ±16% of the patient's maximal inspiratory pressure were performed. Good to excellent agreement (Intraclass correlation coefficients: 0.73–0.97) was observed for all breathing characteristics. When individual differences were plotted against mean values of breaths recorded by both devices, small average biases were observed for all breathing characteristics. To conclude, the training device provides valid assessments of breathing characteristics to quantify inspiratory muscle effort (e.g. work of breathing and peak power) during IMT in patients with weaning difficulties. Availability of valid real-time data of breathing responses provided to both the physical therapist and the patient, can be clinically usefull to optimize the training stimulus. By adapting the external load based on the visual feedback of the training device, respiratory muscle work and power generation during IMT can be maximized during the training.

**Data Availability Statement:** The data used in the present study is deposit within a data repository

which can be accessed via the following link: https://doi.org/10.6084/m9.figshare.c.5499057.

**Funding:** This research project was supported by Research Foundation Flanders (FWO, https://www.fwo.be/en/): grant number G053721N. The funders had no role in study design, data collection and analysis, decision to publish, or preparation of the manuscript.

**Competing interests:** I have read the journal's policy and the authors of this manuscript have the following competing interests: the POWERbreathe KH2 devices, used in this study were provided by HAB - POWERbreathe International Ltd., (Warwickshire, England) for study purposes at no additional cost. This does not alter our adherence to PLOS ONE policies on sharing data and materials.

## Introduction

Inspiratory muscle training (IMT), commonly performed with mechanical threshold loading devices, improves respiratory muscle function and might enhance weaning outcomes in patients with weaning difficulties [1]. These mechanical devices however do not provide real time feedback on breathing characteristics assessed during the training. This would provide valuable information for the physical therapist to optimize the training stimulus and motivate the patient with visual performance feedback. In patients with COPD it has previously been shown that an electronic inspiratory training device provides valid, automatically processed information on breathing characteristics including tidal volume, inspiratory pressure, work of breathing (WOB) and power generated by inspiratory muscles during IMT sessions [2]. In the same setting adherence to and quality of IMT sessions, as reflected by WOB and power generated by inspiratory muscles, were related to improvements in inspiratory muscle function after training [3]. If accurate, the real-time feedback on breathing characteristics during each training session, could offer useful feedback to both patient and physical therapist to optimize the training stimulus during IMT sessions on a breath-by-breath basis in patients with weaning difficulties. As a consequence the interest in performing IMT with an electronic inspiratory loading device which provides real-time data on breathing characteristics in patients weaning from mechanical ventilation is increasing [4–6]. The aim of this study was therefore to investigate the measurement validity of this training device to provide accurate assessments of breathing characteristics during IMT in patients with weaning difficulties. Our hypothesis was that the automatically processed information of the electronic inspiratory training device would agree well with data from a reference measurement device.

## Materials and methods

### Inspiratory muscle training

Patients with weaning difficulties were recruited from an ongoing randomized controlled trial investigating the effects of IMT in patients with weaning difficulties (clinicaltrials.gov identifier: NCT03240263) [4]. Ethical approval was obtained from the responsible local ethics committee (Ethische Commissie Onderzoek UZ/KU Leuven protocol ID: S60516) [4]. Written informed consent was obtained from all patients if awake and alert or from a family member if patients were unable to provide written consent. The consent procedure was approved by the aforementioned local ethics committee and was in line with general data protection regulation (GDPR) inforced by the european union and good clinical research practice guidelines (GCP). The individual in this manuscript has given written informed consent (as outlined in PLOS consent form) to publish these case details (Fig 1). Patients performed daily IMT sessions with the electronic training device (EITD, POWERbreathe KH2, POWERbreathe International Ltd, UK). These sessions consisted of 4 sets of 6–10 breaths against a tapered flow-resistive load [4]. Training intensity was set at 30–50% of their maximal inspiratory pressure (PImax) and adjusted over the course of the training period to the highest tolerable loading. Standardized instructions and encouragements to perform fast and forceful inspirations against the external load followed by complete expirations were provided to the patients [4].

### Measurements and data processing

One training session per week, was randomly chosen for analysis. Airflow and airway pressure measurements were simultaneously sampled and processed at 500Hz by the electronic inspiratory training device and at 75Hz with a portable spirometer (Pocket-Spiro USB/BT 100, M.E. C, Belgium) providing reference data (Fig 1 for more details on the setup of measurements).

A

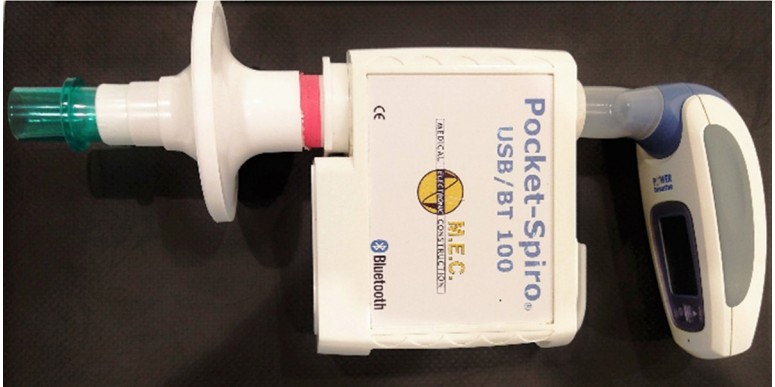

B

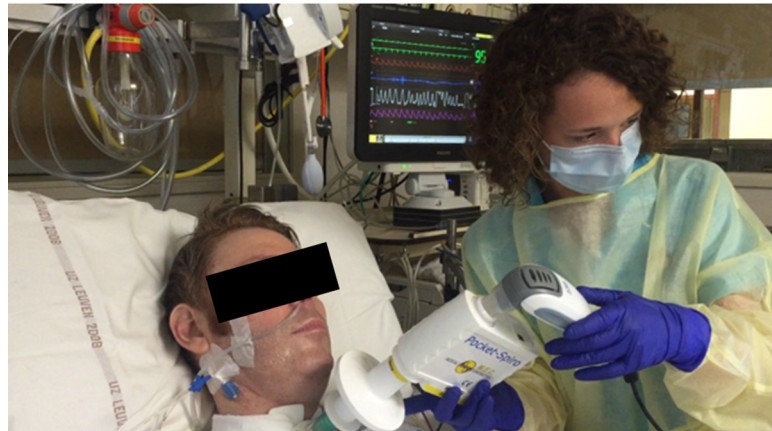

**Fig 1. Setup of measurements.** Inspiratory muscle training was performed with the portable spirometer placed between the electronic inspiratory training device (Panel A) and connected to the patient either via an endotracheal tube or via a tracheostomy (Panel B).

Volume calibration of the portable spirometer was performed daily against the selected external load prior to the IMT session to ensure valid flow and volume measurements. Data sampled by the portable spirometer required semi-manual processing of continuous airway pressure, flow and volume signals with the use of a specific respiratory script in a dedicated software package (Spike 2, version 8.2, Cambridge Electronic Design Limited, United Kingdom). In contrast, data from the training device were processed by a dedicated software package (Breathe-Link software, version 3.3.2a, POWERbreathe International Ltd, UK) and data were available in real time during the sessions for direct feedback, and immediately upon completion of each IMT session for further processing (Fig 2). The initiations of the inspirations and expirations of the data collected by the portable spirometer were marked automatically by the Spike 2 software based on volume peaks and troughs. This corresponded to the point at which the airway pressures generated by the patient changed from positive to negative (inspiration) and from negative to positive (expiration). The inspiration was terminated at the same point as the expiration was initiated. In contrast, the Breathe-Link software identified the inspiratory phase of a breath from the moment that inspiratory airway pressure exceeded

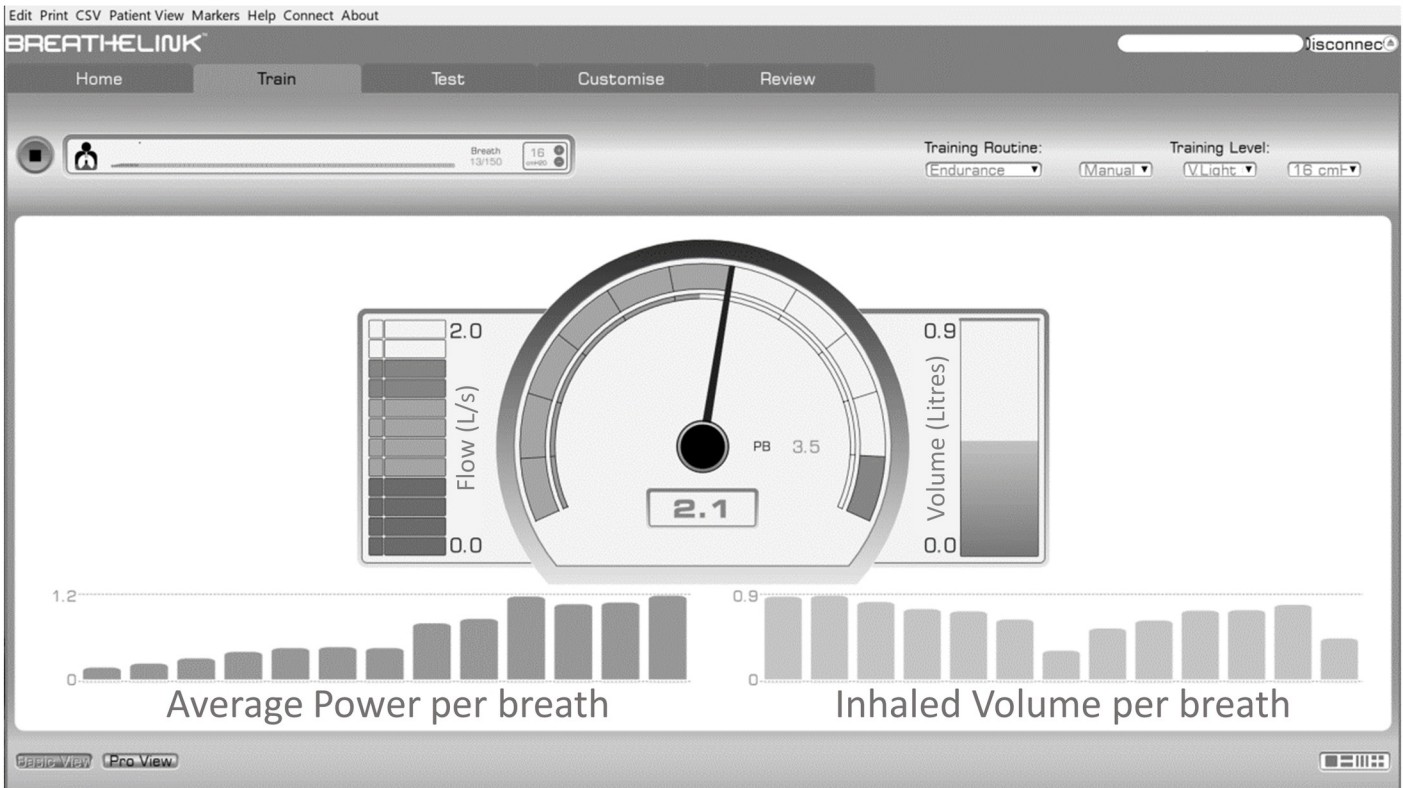

**Fig 2. Visual feedback.** Real-time visual feedback on breathing characteristics are provided by the Breathe-Link software. In the upper left corner, the amount of inspirations that have been performed against the set inspiratory load is indicated. The general settings of the training are depicted in in upper right corner. In the middle of the screen the pointer indicates the peak power that is performed in the last inspiration. On the left of the pointer the peak flow and on the right of the pointer the inspiratory tidal volume of the last inspiration are shown. In the bars below the mean power and the inspiratory tidal volume of all breaths performed in the session are depeicted.

-3cmH$_2$O for at least 10ms and the expiratory phase from the moment that expiratory airway pressure exceeded +3 cmH$_2$O for at least 10ms (information provided by the manufacturer). Additionally, pressure signals from the portable spirometer needed to be smoothed with a time constant of 0.13s to enable the Spike 2 software to achieve a successful computation of work of breathing (see S1 Fig for an example of this smoothing procedure). The Breathe-Link software smooths automatically the airflow and pressures signals of every breath during the IMT session. Both software packages calculated inspiratory work of breathing by integrating the product of pressure and volume of every inspiratory cycle and inspiratory power by integrating the product of pressure and flow of every inspiratory cycle. In some training sessions, the EITD did not detect all breaths that were recorded by the portable spirometer. These breaths were manually identified based on the pairing of breathing characteristics (i.e. tidal volume, mean and peak pressure and mean and peak inspiratory flow) between the data of the EITD and the spirometer of sequential breaths. Discordant breaths recorded by the portable spirometer were identified as missing.

## Data analysis

Intraclass correlation coefficient (ICC) estimates and their 95% confidence intervals were calculated based on breathing characteristics of individual breaths, with a two-way mixed effects model in which patients' effects were regarded as random, measures effects were regarded as

fixed and an absolute agreement definition was applied. ICC values of less than 0.50 were considered as poor agreement, between 0.50 and 0.75 as moderate, between 0.75 and 0.90 as good and above 0.90 as excellent agreement [7]. Individual differences between breaths recorded by the devices (data of the electronic inspiratory training device minus data of the portable spirometer) were plotted against the average scores in Bland-Altman plots. Based on repeatability criteria for spirometry maneuvers a clinically acceptable average bias of less than 0.15L for inspiratory tidal volume and less than 0.30L/s for inspiratory flow were defined [8, 9]. Both ICC and Bland and Altman plots were used to judge the level of agreement between the training device and the portable spirometer.

In the exceptional case the training device did not detect all breaths that were recorded by the portable spirometer and it was not possible to pair the detected breaths with certainty, these training sessions had to be excluded from the ICC analyses and Bland-Altman analyses. To determine the differences in breathing characteristics between the breaths that were detected by both devices and the breaths that were only detected by the handheld spirometer, a backward stepwise logistic regression was performed. Pressure, flow, and volume signals were included in the regression model as explanatory (i.e. independent) variables. To assess the goodness of fit of the model, the Nagelkerke $R2$ and Hosmer–Lemeshow tests were used. To assess the discriminative performance, the area under the curve (AUC) was calculated. Bland-Altman plots were created and AUC were calculated with GraphPad Prism (GraphPad Software, version 9, LCC, United States). Intraclass correlations and backward stepwise logistic regression models were performed with SPSS Statistics (IBM Corp. Released 2019. IBM SPSS Statistics for Windows, Version 26.0. Armonk, NY: IBM Corp).

## Results

One thousand and two breaths of 27 training sessions against an external load of 46 ± 16% of the patient's PImax were analyzed (Table 1). Table 2 provides the results on agreement between

**Table 1. Subject characteristics.**

|                              | n = 13       |
| ---------------------------- | ------------ |
| Sex, male/female, n          | 8/5          |
| Age, y                       | 51 (19)      |
| Body mass, kg                | 66 (18)      |
| Height, cm                   | 167 (9)      |
| PImax, cmH$_2$O              | 36 (13)      |
| PImax, %predicted            | 37 (16)      |
| FVC, L                       | 0.9 (0.4)    |
| FVC, %predicted              | 25 (9)       |
| APACHE-II (0–70)             | 20 (7)       |
| Medical ICU/Surgical ICU, n  | 5/8          |
| ICU length of stay, days     | 68 (46)      |
| Diagnosis, n (%)             |              |
| Double lung transplantation | 5 (39) |
| Pneumonia        | 3 (23)       |
| Hematologic      | 2 (15)       |
| Cardiac failure  | 2 (15)       |
| Thoracic surgery | 1 (8)        |

Values are expressed as mean (standard deviation) or sample size (n) with the corresponding proportion of the total sample size (%). PImax: Maximal inspiratory pressure, FVC: Vital capacity, APACHE-II: Acute physiology and chronic health evaluation; ICU: intensive care unit.

**Table 2. Agreement between the devices.**

| Breathing characteristics | Portable spirometer | | Inspiratory training device | | Intraclass correlation coefficient | | Bland-Altman plot (EITD—Spirometer) | | |
|---|---|---|---|---|---|---|---|---|---|
| n = 809 breaths | Median | (IQR) | Median | (IQR) | ICC | (95% CI) | Bias | (95% LoA) | Rel. bias |
| Mean Pi, cmH$_2$O | 7.0 | (4.7 to 9.9) | 8.9 | (6.3 to 12.6) | 0.90 | (0.29 to 0.96) | 2.1 | (-1.5 to 5.7) | 26% |
| Peak Pi, cmH$_2$O | 13.5 | (8.2 to 19.9) | 14.7 | (9.9 to 22.5) | 0.97 | (0.64 to 0.99) | 2.1 | (-1.4 to 5.6) | 16% |
| Insp. tidal volume, L | 0.47 | (0.32 to 0.63) | 0.44 | (0.27 to 0.62) | 0.88 | (0.86 to 0.89) | -0.04 | (-0.58 to 0.49) | -4% |
| Mean insp. flow, L/s | 0.35 | (0.23 to 0.49) | 0.42 | (0.28 to 0.56) | 0.73 | (0.64 to 0.79) | 0.07 | (-0.30 to 0.43) | 15% |
| Peak insp. flow, L/s | 0.78 | (0.60 to 1.07) | 0.87 | (0.69 to 1.05) | 0.84 | (0.77 to 0.89) | 0.08 | (-0.35 to 0.52) | 12% |
| WOB/breath, Joules | 0.42 | (0.25 to 0.89) | 0.42 | (0.23 to 0.82) | 0.94 | (0.92 to 0.95) | -0.10 | (-0.82 to 0.63) | -8% |
| Mean power, Watts | 0.30 | (0.19 to 0.50) | 0.36 | (0.24 to 0.67) | 0.89 | (0.84 to 0.92) | 0.09 | (-0.36 to 0.53) | 22% |
| Peak power, Watts | 0.85 | (0.52 to 1.53) | 0.94 | (0.61 to 1.86) | 0.96 | (0.93 to 0.97) | 0.14 | (-0.57 to 0.85) | 15% |

Of the 1002 breaths recorded by the portable spirometer, 65 breaths were missed by the electronic inspiratory loading device and were therefore excluded from the analysis. Additionally, 128 breaths of three training sessions contained some breaths that were not recorded by the electronic inspiratory loading device. These could not be unambiguously identified and therefore the entire training sessions were excluded. EITD: electronic inspiratory training device, IQR: Interquartile Range, 95% CI: 95% Confidence Interval, 95% LoA: 95% Limits of Agreement, Pi: inspiratory pressure, Insp.: inspiratory, WOB: Work of Breathing.

the devices. ICCs ranging from 0.73–0.97 and small average biases were observed for the different breathing characteristics, (see Bland Altman plot in Fig 3 and Table 2). The training device did not detect 6% of breaths that were registered by the reference device. The missed breaths were characterized by low tidal volume (below 0.10L) and a low peak inspiratory flow (below a median of 0.36 L/s). (S2 Fig and S1 File). The total WOB per IMT session had a median of 20.42 Joules (interquartile range: 10.00 to 32.72 Joules), while the WOB performed in the breaths that were not detected by the EITD had a median of 0.69 Joules (interquartile range: 0.20 to 1.32 Joules). The share of these breaths in the total WOB was approximately 3% of the total work performed during a training session. The logistic regression explained 37% (Nagelkerke R2) of the variance in missing breaths and correctly classified 94% of the cases ($\chi$2 = 153.030, p<0.001). Sensitivity was 26%, specificity was 98%, positive predictive value was 55% and negative predictive value was 95%, with an area under the curve of 0.89 (95%CI 0.85–0.93, p<0.001). The overall fit of the model was acceptable (Hosmer and Lemeshow goodness of fit test, p = 0.11). Of the five variables included in the regression model, only the inspiratory tidal volume and peak inspiratory flow were significant predictors for missing breaths by the EITD (S1 File for results).

## Discussion

### Main findings

Our data support the validity of an electronic inspiratory muscle training device as a tool to assess breathing characteristics during IMT in patients with weaning difficulties. Good to excellent agreement between the training device and a reference device was observed for all breathing characteristics. The small number of breaths that were not detected by the training device (6%) did not have an important impact on the ability to quantify external work and power generated during the inspiratory muscle training sessions.

### Agreement between devices

For all breathing characteristics small median differences were observed between the devices (Table 2). The difference between the median tidal volume was 0.03L and the difference of the median mean and peak inspiratory flow were 0.07L/s and 0.09L/s. The difference in tidal volume and inspiratory flow were acceptable as these were lower than the predefined acceptable

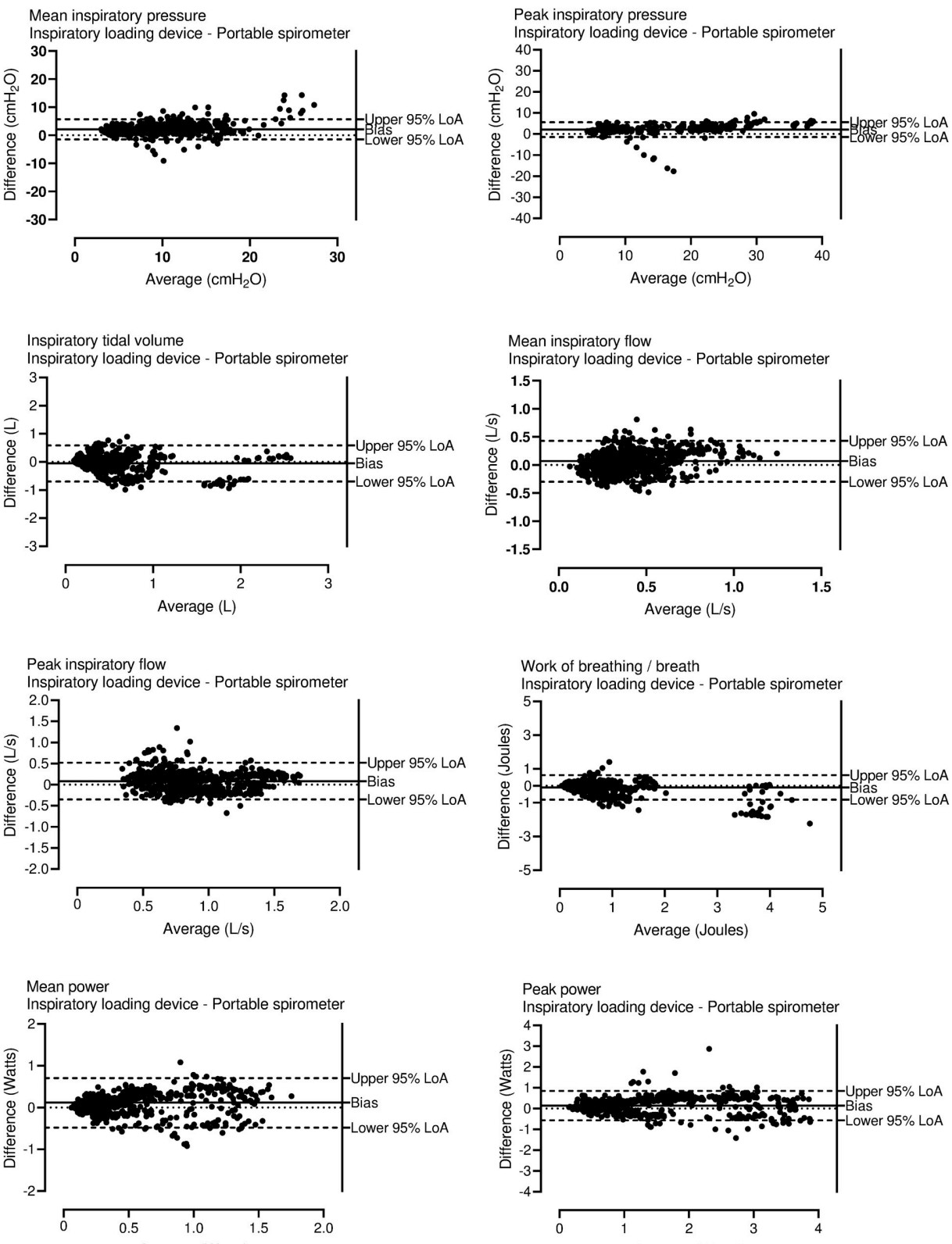

**Fig 3. Breathing characteristics: Bland-Altman plots.** Agreement between estimates of several breathing characteristics of individual breaths performed during inspiratory muscle training as assessed by the portable spirometer and the electronic inspiratory training device. LoA = Limit of Agreement.

maximal differences of 0.15L for tidal volume and 0.30L/s for mean and peak inspiratory flow [8, 9]. High ICCs for all estimated breathing characteristics indicate a good to excellent agreement of the EITD. For peak inspiratory pressure, WOB and peak power an excellent agreement was found (ICC >0.90). These variables are considered crucial to quantify the load on the inspiratory muscles during IMT [3]. No consistent biases were observed for inspiratory tidal volume, WOB per breath, mean and peak inspiratory flow and peak power when plotting average values against individual differences. Although results did vary on an individual breath-by-breath basis, average biases were small supporting the good average agreement between methods (Fig 3 and Table 2).

The validity of the electronic inspiratory training device had already been established in patients with COPD [2]. While the resistive inspiratory load relative to PImax during IMT are similar between patients with COPD (55 ± 13% PImax) and patients with weaning difficulties (46 ± 16% PImax), their PImax was very different (Table 1) [2]. Patients with COPD had a PImax of 67 ± 21 cmH$_2$O (64 ± 18%predicted) while patients with weaning difficulties had a PImax of 36 ± 13 cmH$_2$O (37 ± 16%predicted, Table 1) [2]. Additionally, patients with COPD have very different breathing characteristics during loaded breathing tasks in comparison to patients with weaning difficulties. Mean inspiratory pressure was on average 18.5 cmH$_2$O in the patients with COPD and in the patients with weaning difficulties, the median was 7.0 cmH$_2$O. Mean inspiratory tidal volume for patients with COPD was on average 1.9 L and in patients with weaning difficulties 0.5L, while peak inspiratory flow was on average 2.3 L/s in patients with COPD and patients with weaning difficulties 0.8 L/s. WOB per breath was on average 3.9 J in patients with COPD and 0.42 J in patients with weaning difficulties. Mean power per breath was on average 2.2 Watts in patients with COPD and 0.30 Watts in patients with weaning difficulties [2]. For all breathing characteristics, small and acceptable average biases were observed as they were smaller than the predefined accetablility criteria for tidal volume ($\leq$ 0.15 L) and for inspiratory flow ($\leq$ 0.30 L/s, Table 2) [8, 9]. The biases of the estimates of mean inspiratory pressure (average bias: 2.1 cmH$_2$O) and mean inspiratory power (average bias: 0.09 W, Table 2) were similar to biases observed in previous comparisons in patients with COPD (bias of 0.8 cmH$_2$O and bias of 0.07 W, respectively) [2]. Similar small relative biases for work of breathing were described in both studies, with a relative bias of 5% in patients with COPD and 8% in patients with weaning difficulties [2]. Lower absolute values of the breathing characteristics in patients with weaning difficulties might have resulted in larger relative biases than those previously observed in patients with COPD (Table 2) [2]. The relative bias of EITD's estimates of mean inspiratory pressure (+26%) and mean power (+22%) were larger than those found by Langer and colleagues in patients with COPD (+4% and +3%, respectively) [2].

## Limitations

The highest systematic average biases were found for mean inspiratory pressure and mean power (Table 2). This can be attributed to the difference in definitions of the breathing cycle time applied by the EITD and the portable spirometer as described in the Measurements and data processing section. This shorter inspiratory period detected by the EITD, resulted in systematically higher estimates for mean inspiratory pressure and mean power with the EITD. This interpretation can be extended to the estimates of the mean inspiratory flow, which are also consistently higher with the EITD.

## Clinical implications

Our findings indicate that data on breathing characteristics recorded by the EITD are valid in patients with weaning difficulties. This allows the physical therapist to utilize the automatically

processed and immediately available real-time data provided by the training device to optimize the training stimulus. The accurate feedback on breathing characteristics provided during IMT can be used to titrate the load to the highest tolerable loading and subsequently to maximize the WOB and power during the training session (Fig 2) [3]. In our clinical practice the highest tolerable load is defined as the load that permits a volume expansion of approximately 70% of the vital capacity [4]. Further increasing the load will result in reduced inspiratory volumes, inspiratory flow, work of breathing and power performed during the training. When tidal volume during the training consistently exceeds 70% of the vital capacity, an effort can be made to increase the load to further maximize WOB and power generation during the training, taking also into account the symptoms of dyspnea and breathing effort reported by the patient during the training [4].

In general, it has been shown that verbal and visual feedback during resistive training results in improved training quality and enhances acute improvements in strength and power output and may contribute to improve long-term adaptation in the muscles [10, 11]. Accordingly, in patients with COPD it has been shown that providing verbal feedback on performing maximal inspirations with maximal velocity, in combination with reliable visual feedback on inspired tidal volumes and inspiratory flow during IMT, enhances the patient's motivation and improves the quality of the training session [3]. In patients with COPD it has been shown that higher WOB (higher total training volume) during the training period was associated with larger improvements in inspiratory muscle function [3].

## Conclusions

The training device provides valid data of breathing characteristics to quantify the load on the inspiratory muscles during IMT in patients with weaning difficulties. This provides valuable real-time data to the physical therapist to control and adapt the training stimulus according to the observed breathing responses during the training. By providing visual feedback on breathing characteristics to patients during IMT, the quality of the training sessions, reflected by WOB and power generated by the inspiratory muscles, will be improved. Whether this might result in an additional gain in inspiratory muscle function, such as in patients with COPD, needs further investigation.

## Supporting information

**S1 Fig. Calculation of work of breathing by the use of unsmoothed versus smoothed pressure-volume loops.** Example of the comparison of the calculation of the work of breathing performed on the raw signal of the portable spirometer (Panel A) and the calculation of the smoothed signal with a time constant of 0.13s in the Spike 2 software (Panel B).
(PDF)

**S2 Fig. Comparison between recorded breaths and breaths that were not detected by the inspiratory training device.** Comparison of tidal volume, mean inspiratory flow and peak inspiratory flow. Breaths that were recorded by the portable spirometer (n = 937) were compared to breaths that were recorded by portable spirometer but missed by the inspiratory training device (n = 65) and compared to all the breaths recorded by the inspiratory training device (n = 937). Red line indicates the median. EITD: electronic inspiratory training device.
(PDF)

**S1 File. Logistic regression analysis of breathing characteristics associated with breaths not being detected by the inspiratory training device.**
(DOCX)

## Acknowledgments

We would like to thank all patients who were willing to participate in this study. The authors acknowledge HaB international (Southam, United Kingdom) for providing the training equipment used in the study.

## Author Contributions

**Conceptualization:** Marine Van Hollebeke, Jan Muller, Greet Hermans, Rik Gosselink, Daniel Langer.

**Data curation:** Marine Van Hollebeke, Diego Poddighe, Tin Gojevic.

**Formal analysis:** Marine Van Hollebeke, Diego Poddighe, Tin Gojevic.

**Funding acquisition:** Greet Hermans, Rik Gosselink, Daniel Langer.

**Investigation:** Marine Van Hollebeke, Beatrix Clerckx.

**Methodology:** Marine Van Hollebeke, Diego Poddighe, Beatrix Clerckx, Jan Muller, Greet Hermans, Rik Gosselink, Daniel Langer.

**Project administration:** Marine Van Hollebeke.

**Supervision:** Jan Muller, Greet Hermans, Rik Gosselink, Daniel Langer.

**Visualization:** Marine Van Hollebeke.

**Writing – original draft:** Marine Van Hollebeke, Diego Poddighe.

**Writing – review & editing:** Marine Van Hollebeke, Diego Poddighe, Tin Gojevic, Beatrix Clerckx, Jan Muller, Greet Hermans, Rik Gosselink, Daniel Langer.

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
