## [Decision Letter · Decision Letter 0]

4 Jul 2021

PONE-D-21-15405

Measurement validity of an electronic training device to assess breathing characteristics during inspiratory muscle training in weaning failure patients

PLOS ONE

Dear Dr. Langer,

Thank you for submitting your manuscript to PLOS ONE. After careful consideration, we feel that it has merit but does not fully meet PLOS ONE’s publication criteria as it currently stands. Therefore, we invite you to submit a revised version of the manuscript that addresses the points raised during the review process.

We look forward to receiving your revised manuscript.

Kind regards,

Shane Patman, PhD

Academic Editor

PLOS ONE

Journal Requirements:

3. Please include in your Methods details of the consent procedure. Please include information on whether the IRB specifically approved the consent procedure, and whether it was in line with national regulations.

[I have read the journal's policy and the authors of this manuscript have the following competing interests: the POWERbreathe KH2 devices, used in this study were provided by HAB - POWERbreathe International Ltd., (Warwickshire, England) for study purposes at no additional cost.].

6. We note that Figure 1 B includes an image of a patient and a nurse. 

Reviewers' comments:

Reviewer's Responses to Questions

**Comments to the Author**

1. Is the manuscript technically sound, and do the data support the conclusions?

Reviewer #1: Yes

Reviewer #2: Yes

2. Has the statistical analysis been performed appropriately and rigorously? 

Reviewer #1: Yes

Reviewer #2: Yes

3. Have the authors made all data underlying the findings in their manuscript fully available?

Reviewer #1: Yes

Reviewer #2: Yes

4. Is the manuscript presented in an intelligible fashion and written in standard English?

Reviewer #1: Yes

Reviewer #2: Yes

5. Review Comments to the Author

Reviewer #1: 28-Jun-2021

PLOS ONE

PONE-D-21-15405: Measurement validity of an electronic training device to assess breathing characteristics during inspiratory muscle training in weaning failure patients

This is a manuscript that does not raise a clinical question. It was designed to compare the measurements of selected respiratory parameters obtained by a new electronic device (Powerbreathe KH2, POWERbreathe International Ltd, UK) with the ones obtained more traditionally, using a portable spirometer (Pocket-Spiro USB/BT 100, M.E.C, Belgium). The authors aimed to validate the measures of the new device. The question brought about by the study is relevant considering that it checks if a device with a potentially wide application generates reliable numbers.

The study succeeds in its general purpose, but below are some comments to clarify some points posed in the manuscript.

1. In the title. Weaning failure is more an event than a stable situation. It would be convenient to replace “weaning failure” by “difficult weaning”. In this sense, this change would apply to the whole manuscript.

2. In the abstract. Both devices (the EITD and the spirometer) could have their model and manufacturer mentioned here.

3. Line 135. Replace the word confident por confidence

4. Line 137. Replace the word “patients effects” with “patients' effects”.

5. Table 1. Please replace “VC” by “VT”. Abbreviations “VT” and “PImax” need to be explicated in the table legend.

Reviewer #2: This is an interesting study which concludes that an electronic inspiratory training device (EITD; POWERbreathe KH2) provides valid measurements of breathing characteristics (inspiratory muscle work and peak power) when compared to the gold standard device (Pocket-spiro) during inspiratory muscle training in weaning failure patients. The important clinical implications of this study are clear, and relate to the ability of the EITD to provide breathing characteristics during training in real-time, allowing for therapists to optimise the training stimulus during every training session, and helping motivate the patient via visual feedback.

The manuscript is well written and the analyses were well conducted. My main query with this study relates to the disparity of the sample size within this manuscript and the abstract presented by the authors at the European Respiratory Society (ERS) congress 2020 (included in the European Respiratory Journal proceedings ) entitled “Measurement validity of an electronic inspiratory loading device during inspiratory muscle training in weaning failure patients” [1]. Within the congress abstract there are 30 participants included, with 2594 breaths from 64 training sessions analysed compared to 13 participants and 1002 breaths from 27 training sessions analysed in the current manuscript. The findings of both versions are similar, with an even higher ICC and lower relative bias (% difference) between the devices within the ERS abstract for most variables. However, mean inspiratory resistance during training was much lower in the ERS abstract (24% of maximal inspiratory pressure; PImax) compared to this manuscript (46% PImax). Please could you comment on the reason behind the lower sample size and higher training intensity within this manuscript compared to previously reported results [1]. Were training sessions at a lower inspiratory resistance (i.e. lower % PImax) excluded from the analysis within this manuscript and could this have affected the results in any way?

Specific comments:

Table S1. – For the peak inspiratory pressure logistic regression model coefficient, I think there is a typo - please clarify the value.

1. Van Hollebeke, M., Poddighe, D., Clerckx, B., Hermans, G., Gosselink, R., & Langer, D. (2020). Measurement validity of an electronic inspiratory loading device during inspiratory muscle training in weaning failure patients.

6. PLOS authors have the option to publish the peer review history of their article (what does this mean?). If published, this will include your full peer review and any attached files.

Reviewer #1: No

Reviewer #2: No

---

## [Author Response · Author response to Decision Letter 0]

7 Jul 2021

Response to reviewers and editir comments

Journal Requirements:

Answer: Changes have been made across the whole manuscript to be in accordance to the style requirements of PLOS ONE: 

• Titles of all authors have been removed (Lines 5 and 6)

• Annotation of contact information of the corresponding author has been changed (lines 16-19)

• Author contribution statement has been removed (lines 21-22)

• Subheadings in the abstract have been removed (Lines 24, 33, 39 and 45). 

• Italic text formatting has been removed on line 246, 249 and 250 to be in accordance with the style requirements. 

• The headings “Acknowledgements” and “References” have been changed from a level 2 heading to a level 1 heading according to the style requirements (Line 263 and 269)

• The line spacing in the manuscript has been changed from a 1.5 spacing to a double spacing. 

Answer: The reference list has been checked and is complete and correct: 

Reference 6: Guimarães BDS, de Souza LC, Cordeiro HF, Regis TL, Leite CA, Puga FP, et al. Inspiratory Muscle Training With an Electronic Resistive Loading Device Improves Prolonged Weaning Outcomes in a Randomized Controlled Trial. Critical care medicine. 2020;Publish Ahead of Print. doi: 10.1097/CCM.0000000000004787 is a reference referring to an article published ahead of print and has now been updated (line 290-293)

3. Please include in your Methods details of the consent procedure. Please include information on whether the IRB specifically approved the consent procedure, and whether it was in line with national regulations.

Answer: Information on the consent procedure has been added to the manuscript on lines 79 to 81. 

[I have read the journal's policy and the authors of this manuscript have the following competing interests: the POWERbreathe KH2 devices, used in this study were provided by HAB - POWERbreathe International Ltd., (Warwickshire, England) for study purposes at no additional cost.].

Answer: Hereby, I confirm that the competing interest addressed does not alter our adherence to PLOS ONE policies on sharing data and materials. The updated Competing Interests statement has been added to the cover letter. 

Answer: The data availability statement has been updated in the cover letter including the relevant DOI to access the data. 

6. We note that Figure 1 B includes an image of a patient and a nurse. 

Answer: Statement on the consent for publication of the patient in figure 1B has been added to the method section (line 81-82). The patient included in the image has been informed of the terms of the PLOS open-access license and signed the consent form for publication in a PLOS Journal. The therapist included in the figure is the first author of the current study (Marine Van Hollebeke) and has signed the consent form for publication in a PLOS Journal. 

Response to reviewers

Reviewer #1: 28-Jun-2021

PLOS ONE

PONE-D-21-15405: Measurement validity of an electronic training device to assess breathing characteristics during inspiratory muscle training in weaning failure patients

This is a manuscript that does not raise a clinical question. It was designed to compare the measurements of selected respiratory parameters obtained by a new electronic device (Powerbreathe KH2, POWERbreathe International Ltd, UK) with the ones obtained more traditionally, using a portable spirometer (Pocket-Spiro USB/BT 100, M.E.C, Belgium). The authors aimed to validate the measures of the new device. The question brought about by the study is relevant considering that it checks if a device with a potentially wide application generates reliable numbers.

The study succeeds in its general purpose, but below are some comments to clarify some points posed in the manuscript.

1. In the title. Weaning failure is more an event than a stable situation. It would be convenient to replace “weaning failure” by “difficult weaning”. In this sense, this change would apply to the whole manuscript.

Answer: Thank you for taking the time to read the manuscript and to provide us with valuable feedback. I agree that weaning failure is more an event than a stable condition and that it is appropriate to indicate that we included patients with weaning difficulties. However, I would avoid using the exact term “difficult weaning” as this term has also been used as one of the possible classification groups in the WIND study (1). As we used this classification system and included both patients from the difficult weaning and the prolonged weaning group, I will avoid using “difficult weaning” but instead using the term “weaning difficulties”. Changes have been made in the title (line 1) and across the manuscript (lines: 26, 32, 34, 48, 55, 65, 70, 75, 201, 203, 218, 219, 235, 256) to change ‘weaning failure’ to indicate that patients have ‘weaning difficulties’. 

(1) Beduneau G, Pham T, Schortgen F, Piquilloud L, Zogheib E, Jonas M, et al. Epidemiology of Weaning Outcome according to a New Definition. The WIND Study. Am J Respir Crit Care Med (2017) 195(6):772-83. Epub 2016/09/15. doi: 10.1164/rccm.201602-0320OC. PubMed PMID: 27626706.

2. In the abstract. Both devices (the EITD and the spirometer) could have their model and manufacturer mentioned here.

Answer: On line 36 to 38, the model and manufacturer of the training device and the portable spirometer have been added to the abstract. 

3. Line 135. Replace the word confident por confidence

Answer: Thank you for noticing this error. Changes have been made on line 131

4. Line 137. Replace the word “patients effects” with “patients' effects”.

Answer: Thank you for noticing this error. Changes have been made on line 133

5. Table 1. Please replace “VC” by “VT”. Abbreviations “VT” and “PImax” need to be explicated in the table legend.

Answer: Thank you for this comment and helping to improve the quality of the manuscript. The abbreviations are now explicated in the legend of Table 1. By changing “VC” to “VT”, I assume the purpose would be to indicate that this value represents the tidal volume. I can see that this value might be mistaken for the tidal volume due to the low value (0.9L). However, this value actually represents the forced vital capacity. To clarify this in the table I have changed the abbreviation from “VC” to “FVC”. In addition, I noticed that we have mentioned the abbreviation BMI in the legend but do not provide the value for BMI in the table. Instead we chose to provide body mass and height separately. Therefore, I deleted BMI out of the legend of Table 1. 

Reviewer #2: 

This is an interesting study which concludes that an electronic inspiratory training device (EITD; POWERbreathe KH2) provides valid measurements of breathing characteristics (inspiratory muscle work and peak power) when compared to the gold standard device (Pocket-spiro) during inspiratory muscle training in weaning failure patients. The important clinical implications of this study are clear, and relate to the ability of the EITD to provide breathing characteristics during training in real-time, allowing for therapists to optimise the training stimulus during every training session, and helping motivate the patient via visual feedback.

1. The manuscript is well written and the analyses were well conducted. My main query with this study relates to the disparity of the sample size within this manuscript and the abstract presented by the authors at the European Respiratory Society (ERS) congress 2020 (included in the European Respiratory Journal proceedings ) entitled “Measurement validity of an electronic inspiratory loading device during inspiratory muscle training in weaning failure patients” [1]. Within the congress abstract there are 30 participants included, with 2594 breaths from 64 training sessions analysed compared to 13 participants and 1002 breaths from 27 training sessions analysed in the current manuscript. The findings of both versions are similar, with an even higher ICC and lower relative bias (% difference) between the devices within the ERS abstract for most variables. However, mean inspiratory resistance during training was much lower in the ERS abstract (24% of maximal inspiratory pressure; PImax) compared to this manuscript (46% PImax). Please could you comment on the reason behind the lower sample size and higher training intensity within this manuscript compared to previously reported results [1]. Were training sessions at a lower inspiratory resistance (i.e. lower % PImax) excluded from the analysis within this manuscript and could this have affected the results in any way?

1. Van Hollebeke, M., Poddighe, D., Clerckx, B., Hermans, G., Gosselink, R., & Langer, D. (2020). Measurement validity of an electronic inspiratory loading device during inspiratory muscle training in weaning failure patients.

Answer: Thank you for taking the time to provide us with feedback on the manuscript and for noticing the disparity of the sample size between the manuscript and the abstract previously presented at the ERS congress (1). The patients included in the present study were recruited from an ongoing randomized controlled trial (2). In this RCT, patients were randomized in an intervention group (high-intensity IMT) or a control group (sham low-intensity IMT). In the intervention group, patients performed IMT with a mean training load of 30-50%PImax and in the control group, patients performed IMT with a training load of maximal 10% PImax. In the abstract presented on the ERS congress, we included data from patients from both the intervention group (n=13) and the control group (n=17). 

However, due to valuable feedback during the ERS congress and in consultation with the study team we decided in a later stage to focus the message of the manuscript on the validity of the data on breathing characteristics during IMT of patients randomized in the intervention group only. 

In a clinical setting IMT will be currently performed similarly to the training protocol used in the intervention group (high-intensity IMT). 

Our main aim with the current validity study was to identify whether the IMT device provides valid feedback during IMT to allow the clinician to optimize the training intensity during every training session. Therefore, we focused only on the intervention group, as this is more clinically relevant. Additionally by including both groups, the training intensity is bimodally distributed, as the training intensity is lower than 10% PImax in the control group and between 30 to 50% in the intervention group. This leads to a mean training load of approximately 24%PImax. 

By excluding, the patients randomized in the control group for the present study we aimed improved the clinical message of the validity study. 

(1) Van Hollebeke, M., Poddighe, D., Clerckx, B., Hermans, G., Gosselink, R., & Langer, D. (2020). Measurement validity of an electronic inspiratory loading device during inspiratory muscle training in weaning failure patients.

(2) Hoffman M, Van Hollebeke M, Clerckx B, Muller J, Louvaris Z, Gosselink R, et al. Can inspiratory muscle training improve weaning outcomes in difficult to wean patients? A protocol for a randomised controlled trial (IMweanT study). BMJ Open (2018) 8(6):e021091. Epub 2018/07/02. doi: 10.1136/bmjopen-2017-021091. PubMed PMID: 29961023; PubMed Central PMCID: PMCPMC6042566.

2. Specific comments:

Table S1. – For the peak inspiratory pressure logistic regression model coefficient, I think there is a typo - please clarify the value.

Answer: Thank you for noticing this error. Changes have been made in table S1. The error in the logistic regression model coefficient of the peak inspiratory pressure has been changed from -0..3 to -0.03.

---

## [Decision Letter · Decision Letter 1]

16 Jul 2021

Measurement validity of an electronic training device to assess breathing characteristics during inspiratory muscle training in weaning failure patients

PONE-D-21-15405R1

Dear Dr. Langer,

We’re pleased to inform you that your manuscript has been judged scientifically suitable for publication and will be formally accepted for publication once it meets all outstanding technical requirements.

Kind regards,

Shane Patman, PhD

Academic Editor

PLOS ONE

Additional Editor Comments (optional):

Reviewers' comments:

Reviewer's Responses to Questions

**Comments to the Author**

1. If the authors have adequately addressed your comments raised in a previous round of review and you feel that this manuscript is now acceptable for publication, you may indicate that here to bypass the “Comments to the Author” section, enter your conflict of interest statement in the “Confidential to Editor” section, and submit your "Accept" recommendation.

Reviewer #2: All comments have been addressed

2. Is the manuscript technically sound, and do the data support the conclusions?

Reviewer #2: Yes

3. Has the statistical analysis been performed appropriately and rigorously? 

Reviewer #2: Yes

4. Have the authors made all data underlying the findings in their manuscript fully available?

Reviewer #2: Yes

5. Is the manuscript presented in an intelligible fashion and written in standard English?

Reviewer #2: Yes

6. Review Comments to the Author

Reviewer #2: Thank you for clarifying the reason behind the disparity in sample sizes. I agree that by focusing the validation of the device in an intervention group only, the clinical implications of this study are improved. I have no further suggestions.

7. PLOS authors have the option to publish the peer review history of their article (what does this mean?). If published, this will include your full peer review and any attached files.

Reviewer #2: No

---

## [Editor Report · Acceptance letter]

18 Aug 2021

PONE-D-21-15405R1 

Measurement validity of an electronic training device to assess breathing characteristics during inspiratory muscle training in patients with weaning difficulties 

Dear Dr. Langer:

I'm pleased to inform you that your manuscript has been deemed suitable for publication in PLOS ONE. Congratulations! Your manuscript is now with our production department. 

Kind regards, 

on behalf of

Assoc Prof Shane Patman 

Academic Editor

PLOS ONE